# Post-Translational Modification of ZEB Family Members in Cancer Progression

**DOI:** 10.3390/ijms232315127

**Published:** 2022-12-01

**Authors:** Mi Kyung Park, Ho Lee, Chang Hoon Lee

**Affiliations:** 1Department of Cancer Biomedical Science, Graduate School of Cancer Science and Policy, National Cancer Center, Goyang-si 10408, Republic of Korea; 2College of Pharmacy, Dongguk University, Seoul 04620, Republic of Korea

**Keywords:** zinc finger E-box binding homeobox (ZEB), post-translational modifications (PTM), cancer progression

## Abstract

Post-translational modification (PTM), the essential regulatory mechanisms of proteins, play essential roles in physiological and pathological processes. In addition, PTM functions in tumour development and progression. Zinc finger E-box binding homeobox (ZEB) family homeodomain transcription factors, such as ZEB1 and ZEB2, play a pivotal role in tumour progression and metastasis by induction epithelial-mesenchymal transition (EMT), with activation of stem cell traits, immune evasion and epigenetic reprogramming. However, the relationship between ZEB family members’ post-translational modification (PTM) and tumourigenesis remains largely unknown. Therefore, we focussed on the PTM of ZEBs and potential therapeutic approaches in cancer progression. This review provides an overview of the diverse functions of ZEBs in cancer and the mechanisms and therapeutic implications that target ZEB family members’ PTMs.

## 1. Introduction

Cancer-associated mortality represents the second leading cause of death worldwide after cardiovascular disease [1]. Cancer metastasis is the primary cause of cancer mortality, accounting for approximately 90% of tumour-related deaths. The epithelial-mesenchymal transition (EMT) is the tissue repair and developmental process, along with neural crest formation, heart morphogenesis, and mesoderm formation, facilitating gastrulation and secondary palate formation [2,3,4,5]. Moreover, EMT is a vital clue to tumour invasion and metastasis. Zinc finger E-box binding homeobox transcription factors (ZEBs) play a crucial role in the progression and metastasis of various cancers, as EMT-related transcription factors [6,7,8,9,10,11,12,13], in the regulation of DNA damage repair [14] and neuronal differentiation [15].

Furthermore, ZEBs are associated with the degree of malignancy in various types of cancer and the activation of EMT signalling, which are widely believed to contribute to invasion, metastasis, recurrence and therapeutic resistance. ZEBs are also associated with cancer transformation and EMT. Post-translational modification (PTM) is the enzymatic modification of proteins after synthesis [16] and induces proliferation in cancer progression by regulating the cell cycle, cell survival and cellular signalling [17]. In addition, it is found through structural studies and biochemical studies that cofactors present through covalent bonds at the active sites of enzymes also undergo the PTM process [18,19,20]. Therefore, the PTM of proteins plays a regulatory role in cancer initiation and progression. In turn, the ZEBs are regulated by PTM, including phosphorylation, SUMOylation, ubiquitination, acetylation and deacetylation. This review focuses on the specialised roles of various ZEBs’ PTMs on cancer progression.

## 2. ZEB1 and ZEB2 Proteins and Their Physiological Functions

The zinc finger E-box-binding homeobox 1 (ZEB1) is also known as δEF1, ZFHX1A, MEB1, Nil-2-a, TCF8, AREB6, ZFHEP1 or BZP [21]. The human *ZEB1* gene is located on chromosome 10p11.22 and encodes the 1117 amino acid ZEB1 protein [22]. Zinc finger E-box-binding homeobox 2 (ZEB2) is identified as KIAA0569, SIP1, ZFHX1B and ZFX1B; the human *ZEB2* gene is located on chromosome 2q22.3 and encodes a 1214 amino acid protein [23]. The ZEB proteins consist of a homeodomain (HD) in the middle of the structure and other protein binding domains, including the SMAD interaction domain (SID), which regulates the transforming growth factor beta (TGFβ)-mediated transcription with bone morphogenetic proteins (BMP) signalling, zinc finger domain (ZFD), coactivator binding domain (CBD), CtBP interaction domain (CID) and the p300-CBP-associated factor (P/CAF) binding domain, which control EMT as a trigger of for tumour progression and metastasis (Figure 1) [24,25,26,27,28].

ZEB1 can recruit cosuppressors or coactivators by high-affinity binding of the ZFD to specific DNA binding sites (CACCTG), upregulating or downregulating its target genes [29]. ZEB proteins bind to SMADs. However, while ZEB-1/dEF1 synergises with SMAD proteins to activate transcription, promote osteoblastic differentiation and induce cell growth arrest, ZEB1 is expressed during development in the central nervous system, heart, skeletal muscle and haematopoietic cells; this plays pivotal roles in regulating development, differentiation and maintenance [26,30].

Additionally, ZEB1 is a transcriptional activator, or, has repressor functions in normal regulatory processes and dysregulated progress, such as cancer progression and metastasis. ZEB2 is expressed during the development in the neural tube and crest cells and all parts of the developing forebrain. In addition, it plays a role as a regulator of the TGFβ/BMP signal pathway. When the TGFβ/BMP factor binds to the receptor, the SMAD proteins are translocated to the nucleus, activating the target genes’ transcription. ZEB2 interacts with R-SMADs to induce embryo neutralisation and disrupts the expression of the activin-dependent *Brachyury* gene in *Xenopus* [31,32]. ZEB2 also endures post-transcriptional regulation by several micro-RNAs (miRNAs), such as postnatal brain miRNA (miR9) [33].

## 3. ZEB1 and ZEB2 in Cancer Progression

ZEB protein is involved in tumour invasion and metastasis in the invasive front of carcinomas by EMT induction. ZEB1 is highly expressed in several tumours, including breast [6,7], pancreatic [9,27,34], colorectal [35], gastric [36,37], lung [38,39,40], uterine [41], hepatocellular carcinoma [42], prostate [43,44] and lymphoma [45] cancers. In these tumours, ZEB1 expression correlates with the loss of E-cadherin and is associated with advanced disease or metastasis, indicating the relevance of ZEB1 induction of EMT and tumour progression [13]. Mechanistically, TGF-β enhances pSMAD2/3 and ZEB1 [46] and ZEB2 [47] expression to increase tumour invasion. The β-catenin translocates into the nucleus to activate ZEB1 [48] transcription. WNT signaling induces ZEB2 expression in tumour metastasis [49]. Activation of MEK1/2 and ERK1/2 promotes tumour progression by ZEB1 [50] and ZEB2 [51]. TNF-α induces the mesenchymal phenotype via NF-κB, ZEB1 and ZEB2 signaling [52]. Fos-related antigen 1 (Fra-1) is a member of the Fos family that dimerizes with Jun proteins to form AP-1. Fra-1 induces EMT by modulating ZEB1, ZEB2 and TGFβ expression [53]. E2F1, a transcription factor, regulates EMT and metastasis by increasing ZEB2 expression in small-cell lung cancer [54]. ZEB2 is coexpressed with the POU family and upregulates EMT induction [55]. PRC2-mediated ZEB2 expression represses PTM by SUMOlation [56]. FOXO1, a member of the FOXO family of transcription factors (FoxOs), binds the ZEB2 promoter and destabilizes the ZEB2 mRNA. As a result, it inhibits ZEB2-induced EMT [57] (Figure 2). Loss of E-cadherin is a casual prerequisite for progressing from adenocarcinoma to invasive carcinomas by genetic and epigenetic mechanisms during malignant transformation [8]. In analogy with their function, ZEB1 lose the epithelial phenotype and gain the mesenchymal phenotype with motile and migratory abilities in cancer [5]. Moreover, ZEB1/miR-200 plays an essential role in embryonic development and malignant tumour progression [58]. ZEB1 is an essential factor in the regulation of the initiation and development of tumours through EMT (Figure 3).

In the genetically engineered mouse model (GEMM), *ZEB1* knockout mice die perinatally, exhibiting respiratory failure; severe T cell deficiency of the thymus; and various skeletal defects, including craniofacial abnormalities, limb and sternum defects, and malformed ribs [59]. These developmental defects are associated with mesenchymal-epithelial transition, as evidenced by the re-expression of E-cadherin and loss of vimentin in several tissues and embryonic fibroblasts [60]. In addition, ZEB1 is a crucial factor for local invasion, colonisation capacities and distant metastasis in the Pdx1-Cre-mediated mutant KRAS and the p53 pancreatic cancer mouse (KPC) model [9]. ZEB1 was also shown to affect p53 and RB-dependent oncosuppressive pathways and to prevent senescence and apoptosis, two critical barriers against tumour development. In line with this notion, mouse embryonic fibroblasts (MEF) from *ZEB1* knockout mice undergo early replicative senescence.

ZEB2 is expressed in several tumours, including metastatic ovarian and breast carcinoma [61], pancreatic cancer [62], oral squamous cell carcinomas [63], gastric cancer [64], bladder cancer [65] and glioma [66]. The expressed ZEB2 in tumours is involved in the cell cycle, apoptosis, unregulated cell proliferation, EMT and cancer development and progression [61,64,65,66,67,68]. As T-bet or T-box protein expressed in T cells (TBX21) expression increases, ZEB2 is induced in natural killer (NK) cells. Interaction of ZEB2 and T-bet is required for NK cell maturation to suppress lung melanoma [69]. AKT promotes ZEB2 expression through the nuclear factor-kappa-light-chain-enhancer of activated B cells (NF-κB) pathway in squamous cell carcinoma (SCC) lines [70].

ZEBs also regulate immune checkpoints, evading immune destruction in tumour progression and the microenvironment [71,72]. Programmed death protein 1 (PD-1, encoded by the *PDCD1* gene) and its ligand programmed death-ligand 1 (PD-L1, encoded by the *CD274* gene) function in the immunotherapy of cancer by evading T cell immunity [73]. ZEB1-induced PD-L1 is highly expressed in lung cancer cells [74]. SNHG14/miR-5590-3p/ZEB1 promotes B cell lymphoma progression and evades immunity evasion by exerting PD-1/PD-L1 [75]. ZEB1 is a key target of the melanoma immune escape [76]. ZEB-1 and miR-200 upregulated PD-L1 in breast cancer [77] (Figure 3).

## 4. Post-Translational Modifications of ZEBs in Cancer Progression

PTMs are covalent modifications that occur after the transcript has been translated into proteins, such as the ZEB1, ZEB2, SNAIL (*SNAI1*), SLUG (*SNAI*2) and twist-related (Twist 1) proteins. The human *SNAI1* is located on chromosome 20q13.13 and encodes the 264-amino acid Snail protein. It is a member of the Snail superfamily, and acts as a transcriptional regulator of EMT [78]. The human *SNAI*2 is located on chromosome 8q11.21 and encodes the 268-amino acid Slug protein. Slug binds the nuclear receptor corepressor (NCoR) and C-terminal binding protein 1 (CtBP1) to stabilize Slug and inhibit the expression of E-cadherin [79]. The human *Twist1* genes are located on chromosome 7p21.2 and encodes the 202-amino acid Twist1 protein. The Twist1 plays a critical role in the progression of cancer by modulating EMT [80,81]. These covalent modifications include adding a modifying chemical group or another small protein to one or more residues of the target protein [82]. PTM can occur within the protein on single or multiple residues, undergoing the same or different modifications [83]. Table 1 provides an overview of the molecular mechanisms and biological functions of PTMs of ZEBs in cancer progression.

### 4.1. Phosphorylation

Protein phosphorylation is the most common PTM and is essential for regulating multiple molecular pathways involved in processes such as metabolism, transcription, differentiation and apoptosis. Protein kinases (PK) catalyse phosphorylation by promoting the transfer of ATP’s γ-phosphate to serine, threonine or tyrosine residues on the target proteins. Protein phosphatases (PP) catalyse the reverse process [96,97].

Phosphorylation is a PTM that is known to control ZEB1. Zhang et al. [14] showed that the EMT regulator ZEB1 promotes DNA damage response (DDR) and tumour radioresistance. This regulation is initiated by the phosphorylation and stabilisation of ZEB1 by ATM serine-threonine kinase and is mediated by checkpoint kinase 1 (CHK1) stabilisation by a ZEB1-interacting deubiquitylase, USP7 [14]. Moreover, various CHK1 inhibitors have been tested in anti-cancer clinical trials and warrant investigation as candidate radiosensitising agents for breast tumours with high levels of ZEB1 [98].

Tyrosine kinase receptors related to ZEB1 can also activate EMT. The disassembly of tight junctions during EMT can also be SMAD-independent. TGFβRI and Par6 coexist in tight junctions. On stimulation of TGFβ1, the TβRI-TβRII hetero-dimerise results in a complex containing TβRII/TβRI and Par6 in each tight junction. This interaction results in the phosphorylation of Par6 at Ser345, which is mediated by TβRII [99]. Phosphorylated Par6 interacts with the E3-ubiquitin ligase SMURF-1 that targets RhoA for degradation, leading to the disassembly of tight junctions. Therefore, TGFβ signalling has good therapeutic value [100,101]. GSK3β phosphorylates ZEB2 at Ser705 and Tyr802 in colorectal cancer; this induces EMT, colorectal cancer-cancer stem-like cell properties and metastasis [93].

### 4.2. SUMOylation

SUMOylation is another PTM characterised by the reversible binding of a small ubiquitin-like modifier (SUMO) to the target protein. The three-dimensional structure of SUMO is similar to ubiquitin [102,103]. SUMO modulates DNA damage repair [104,105,106], immune responses, carcinogenesis, cell cycle progression [102,107] and apoptosis. Therefore, SUMOylation is attributed to cancer progression and might function as an actional therapeutic target for cancer [108]. Olig2 SUMOylation has been demonstrated to protect against the genotoxic damage response by antagonising p53 gene targeting [109]. Furthermore, the EMT-related transcription factor ZEB1 was SUMO-modified, and its levels decreased in Senp1-silenced HCC cells [85].

Members of the miR-200 family act as tumour-suppressive miRNAs, enhancing the expression of E-cadherin and suppressing the expression of ZEB1 and ZEB2. The overexpression of miR-200 results in a reduced expression of ZEB transcription factors and enhanced expression of epithelial makers [21,58,110]. In pancreatic cancer, FoxM1 is overexpressed and promotes EMT by the up-regulation of mesenchymal cell markers, such as ZEB1, ZEB2, SLUG and vimentin [111]. Pc2-mediated SUMOylation of ZEB2 (Lys391, Lys866) modulates the repression of E-cadherin transcription [56].

### 4.3. Ubiquitination and Deubiquitination

Ubiquitination regulates the activity and levels of proteins. It is associated with various disease such as cancer, autoimmunity and inflammation [112,113]. Therefore, understanding of ubiquitination processes could provide opportunities for the development of new therapeutics [114]. The seven-in-absentia (Siah) ubiquitin ligases have been reported to decrease the stability of the ZEB1 protein through the ubiquitin-proteasome pathway; this subsequently affects the EMT process in mammalian cancer cells [86]. Deubiquitinating enzymes (DUBs) are key components of the ubiquitin-proteasome system (UPS) that remove ubiquitin chains from their protein substrates [115]. A large body of evidence suggests that the dysfunction of DUBs is responsible for many pathologies, including cancer [116,117]. More recently, there have been reports of associations between DUBs and metastasis in various cancer types. For example, the ectopic expression of ubiquitin-specific protease 14 (USP14) is associated with liver and lymph node metastasis in colorectal cancer [118]. Multiple deubiquitinating enzymes, including USP26, ubiquitin thioesterase (OTUB1) and pMSD3, have been shown to promote the metastasis of oesophageal squamous cell carcinoma through the stabilisation of Snail, which is another EMT transcription factor [119,120,121]. Notably, WP1130 is a partially selective inhibitor of several DUBs, including USP9X, USP5, USP14 and UCH37, and has been shown to trigger a rapid accumulation of polyubiquitinated proteins into aggresomes and induce breast tumour regression [122]; this suggests a potential role for DUBs as therapeutic targets in the treatment of cancer metastasis. Zhang et al. [123] reported that the CDK4/6-USP51-ZEB1 axis plays a vital role in breast cancer metastasis and could be a viable therapeutic target for treating advanced human cancers. In particular, USP51 deubiquitinates ZEB1’s *N*-terminal region and minces metastasis and therapy resistance in breast cancer [87]. USP18 also deubiquitinates ZEB1 and enhances EMT and metastasis in oesophageal squamous cell carcinomas. Furthermore, CSN5 regulates the levels of ZEB1 ubiquitination in renal cell carcinoma and could be a potential therapeutic target [88]. The SBD domain of ZEB2 is required for ubiquitination mediated by Fbxo45 and K48-linkage poly-ubiquitination on ZEB2 [94].

### 4.4. Acetylation and Deacetylation

Acetylation and deacetylation play roles in the regulation of transcriptional activation, nuclear localisation and DNA binding. In addition, they are associated with signalling pathways, cell cycle and ZEB1 at Lys741, LysK774 and Lys775 was acetylated by P/CAF [25]. Tip60, a cell-type-specific transcriptional regulator, acetylates the N-terminal of ZEB1 [90]. ZEB1 recruits the histone deacetylases HDAC1 and HDAC2, which inhibit E-cadherin expression in pancreatic cancer [91].

## 5. Targeting ZEB Modification for Cancer Therapy and Therapeutic Resistance

ZEB is highly expressed in several cancers. Inhibiting the biological function of ZEB is a new approach in cancer therapy. The expression, stability, and localization of ZEB could interfere with different interactions in cancer patients. Phosphorylation of retinoblastoma (Rb) enhances the progression and metastasis of several cancers. The mechanism of RB in cancer is associated with the regulation of cyclins and cyclin-dependent kinase. Rb dephosphorylation regulates EMT by inhibiting the ZEB1 transcriptional activity in breast cancer [124]. There are ZEB PTMs in resistance to various anticancer therapies [125]. The expression level of ZEB1 by induction ATM activation occurs in chemoresistance in human breast cancer [126]. ZEB1 regulates tumor radioresistance by deubiquitylation of USP7. This in turn upregulates homologous recombination-dependent DNA damage repair (DDR) and tumor radioresistance [14]. ZEB1 also modulates the radiotherapy resistance by the inhibition miR-205 [127].

## 6. Summary and Perspective

Highly expressed ZEBs are significantly associated with the tumourigenesis and metastasis of various cancers. Furthermore, the ZEBs, including ZEB1 and ZEB2, are transcription factors that control EMT in tumour progression; this is a process in which tumour cells within the primary tumour lose their cell junctions and their epithelial morphology changes to fibroblastoid morphology. Therefore, understanding the mechanism of ZEBs’ contribution to metastasis is paramount in improving cancer treatment outcomes. However, very little research has been done to target PTMs of ZEBs in cancers. We hypothesised that identifying PTM inhibitors represents an underexplored area of research that may hold significant potential in developing future cancer treatments. Furthermore, identifying ZEBs’ post-transcriptional and PTMs is vital, given that these changes could be identified in the primary tumour before metastasis occurs. Such knowledge can allow clinicians to better predict the patients who have genotypes more likely to follow an aggressive clinical course prone to the development of metastases. These patients could then be treated with different approaches from the onset of the disease to reduce the risk of metastasis, allow for better prognoses and ultimately, enhance survival.

We highlighted the current understanding of the regulation and underlying molecular mechanisms of the ZEB family members’ PTMs in cancer progression; this may provide new insights into developing novel cancer therapeutic strategies and opportunities.

## Figures and Tables

**Figure 1 ijms-23-15127-f001:**
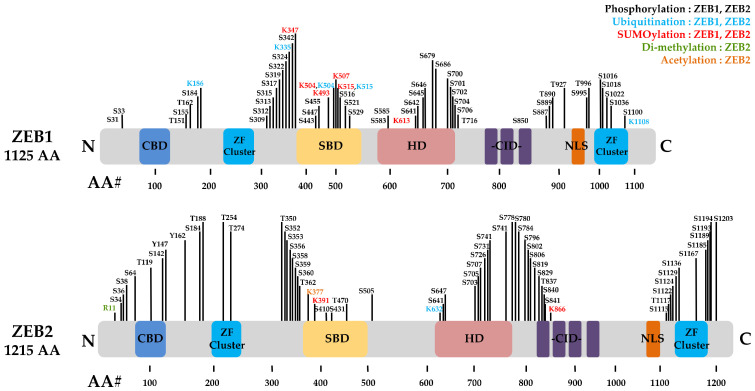
Overviews of ZEB1 and ZEB2 PTMs. It is characterized by the presence of two zinc finger clusters, one at each end (NZF and CZF) and located homeodomain (HD). Other domains are P300-P/CAF interaction domain (CBD), the Smad binding domain (SBD) and the CtBP interaction domain (CID). ZF, zinc finger; NLS, nuclear localization signal. PTM site. Black, Phosphorylation (ZEB1, ZEB2); Sky blue, Ubiquitination (ZEB1, ZEB2); Red, SUMOylation (ZEB1, ZEB2); Green, Di-methylation (ZEB1, ZEB2); Orange, Acetylation (ZEB2).

**Figure 2 ijms-23-15127-f002:**
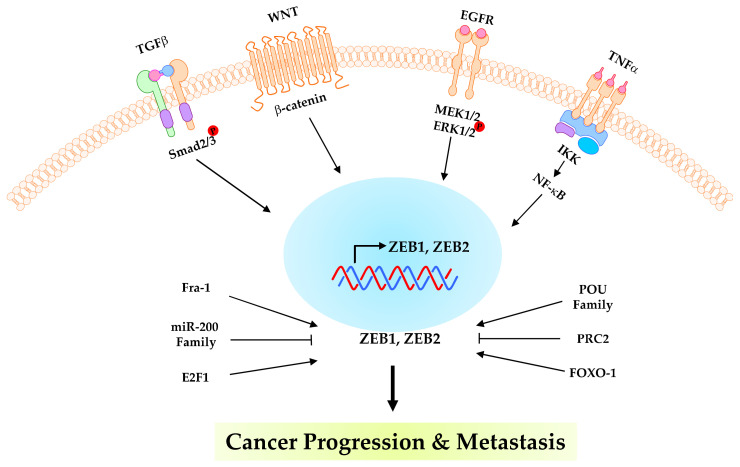
Mechanisms of ZEB family in cancer progression and metastasis. EGFR, WNT, tumor necrosis factor-a (TNFa), transforming growth factor beta (TGF-b), Fos-related antigen 1 (Fra-1), miR-200 family, POU family, PRC2 and FOXO-1 trigger expression of ZEB1 proteins. As a result, the ZEB family controls cancer progression and metastasis.

**Figure 3 ijms-23-15127-f003:**
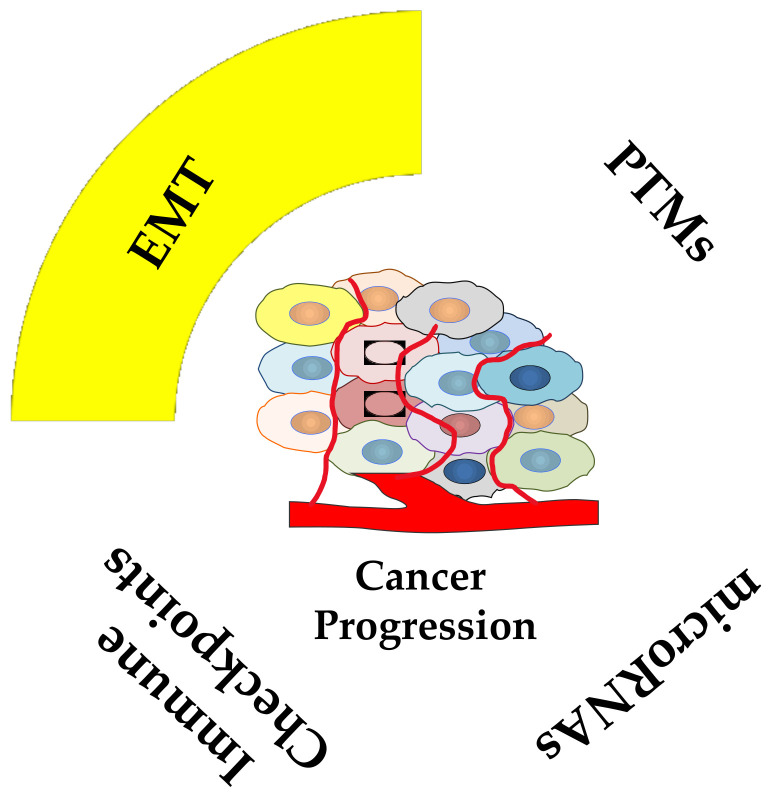
Regulation of ZEB family in cancer progression. PTMs, EMT, miRNA and immune checkpoints of ZEBs functionally are linked to cancer progression.

**Table 1 ijms-23-15127-t001:** Functions of ZEBs-PTMs.

PTMs Type	PTM Sites	Kinase/Enzyme	Biological Function	Cancer Type	Ref.
ZEB1					
Phosphorylation	Thr867	ERK	Inhibition of the nuclear localisation of ZEB1	-	[84]
	Thr851, Ser852, Ser853	PKC	Inhibition of the nuclear localisation of ZEB1	-	[84]
	Ser585	ATM	Promotes DDR and tumour radioresistance	BC	[14]
SUMOylation	-	Senp1	Promotes migration and EMT.	HCC	[85]
Ubiquitination	-	Siah	Promotes cell proliferation and invasion	BC	[86]
Deubiquitination	N-terminal	USP51	Promotes cell proliferation and invasion	BC	[87]
		CSN5	Promotes metastasis and EMT	RCC	[88]
		USP18	Promotes EMT	ESCC	[89]
Acetylation	Lys741, Lys774, Lys775	P/CAF	Promotes the formation of a p300-SMAD transcriptional complex		[25]
	N-terminal	TIP60	Corepressor of the ZEB	T lymphoma	[90]
Deacetylation		HDAC1/2	Promotes EMT	PAAD	[91,92]
ZEB2					
Phosphorylation	Ser705, Tyr802	GSK-3β	Promotes metastasis and chemoresistance	CRC	[93]
SUMOylation	Lys391, Lys866	Pc2	Promotes EMT		[56]
Ubiquitination	Lys48	FBXO45	Promotes EMT initiation and cancer progression		[94]
		FBXL14	Promotes EMT	COAD	[95]
		FBXW7	Promotes metastasis and chemoresistance	CRC	[93]

BC, breast cancer; HCC, hepatocellular carcinoma; RCC, renal cell carcinoma; ESCC, oesophageal squamous cell carcinomas; CESC, cervical squamous cell carcinoma and endocervical adenocarcinoma; PAAD, pancreatic adenocarcinoma; CRC, colorectal cancer; COAD, colon adenocarcinoma; ERK, extracellular signal-regulated kinase; PKC, protein kinase C; ATM, ataxia–telangiectasia mutated kinase; USP51, ubiquitin-specific peptidase 51; CSN5, COP9 signalosome subunit 5; USP18, ubiquitin-specific peptidase 18; PCAF, p300/CBP-associated factor; TIP60, tat-interacting protein of 60 kDa; HDAC1/2, histone deacetylase 1/2; GSK3β, glycogen synthase kinase 3 beta; FBXO45, F-box only protein 45; FBXL14, F-Box and leucine-rich repeat protein 14; FBXW7, F-box/WD repeat-containing protein 7.

## Data Availability

Not applicable.

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
