# Peer review of "Post-Translational Modification of ZEB Family Members in Cancer Progression"

_ijms, 2022, doi:10.3390/ijms232315127_

Round 1
Reviewer 1 Report
Park and co-authors submitt a review on post-translation modifications (PTMs) on ZEB (zinc finger E-box binding homeobox) proteins. ZEB proteins are involvement in cancer metastasis and are associated with the acquisition of cancer and therapy resistance. There is clear scientific merit in revieweing proteins with such characteristicas and in choosing PTM as the motivator to develop the manuscript. However, in its present stage, the manuscript presents deficiences that should be revised to reach the possibilty of publication.
At the end of the text, the authors state that “We highligthed…molecular mechanisms…”. I disagree, no mechanisms were presented, the manuscript presents a survey of literature data, with only modest critical discussion. Actually, addition of chemical mechanism would be a plus.
To make matters worse, most of the text is not well written as most of the paragraphs were built with loose sentences and little effort to link these sentences and improve written.
Page 1, Lines 36-38. PTMs are usually enzymatic because, for instance, co-factors binding are also considered PTMs. The authors should accommodate this information when explaining PTMs.
Page 3 lines 123-125. Several proteins are listed but not explained.
Fig. 1: Very small with low resolution (compare to fig. 2). Top: the meaning of the legends are not explained. It is probably my copy but colors do not match.
Page 5 lines 178-181 are in red.
Table 1. Whats is the meaning of “Regulation factor”? It may be better explained in the text.
Wold it be possible to add a section with a good discussion about medical treatments, in tests or already being applied, that target PTMs in ZEBs?
A ZEB1/2 amino acid alignment would be a nice addition to aid the reader follow the sites of PTMs.
Author Response
[To reviewer 1]
Dear Professor,
We are grateful for your consideration of our manuscript entitled “Post-translational modification of ZEB family members in cancer progression” (manuscript ID: ijms-2031734) by Park et al. and appreciate your helpful comments.
Replies to the comments are as follows:
General Comment: At the end of the text, the authors state that “We highligthed…molecular mechanisms…”. I disagree, no mechanisms were presented, the manuscript presents a survey of literature data, with only modest critical discussion. Actually, addition of chemical mechanism would be a plus.
Reply: Thank you for your comment. We have revised the text and add figure according to your comments. Red characters were used to highlight improved parts of the revised manuscript.
Figure 2. Mechanisms of ZEB family in cancer progression and metastasis. EGFR, WNT, tumor necrosis factor-a (TNFa), transforming growth factor beta (TGF-b), Fos‐related antigen 1 (Fra‐1), miR-200 family, POU family, PRC2, and FOXO-1 trigger expression of ZEB1 proteins. As a result, ZEB family control cancer progression and metastasis.
Mechanistically, TGF-b enhances pSMAD2/3 and ZEB1[46] and ZEB2[47] expression to increase tumour invasion. The b-catenin translocate into nucleus to activate ZEB1[48] transcription. WNT signaling induce ZEB2 expression in tumour metastasis[49]. Activation of MEK1/2 and ERK1/2 promotes tumour progression by ZEB1[50] and ZEB2[51]. TNF-a induces the mesenchymal phenotype via NF-kB, ZEB1 and ZEB2 signaling[52]. Fos‐related antigen 1 (Fra‐1) is a member of Fos family that dimerizes with Jun proteins to form AP‐1. Fra-1 induces EMT by modulating ZEB1, ZEB2 and TGFb expression[53]. E2F1, a transcription factor, regulates EMT and metastasis by increasing ZEB2 expression in small cell lung cancer[54]. ZEB2 is coexpressed with POU family, upregulates EMT induction[55]. PRC2 mediated ZEB2 expression repress PTM by SUMOlation[56]. FOXO1, a member of the FOXO family of transcription factors (FoxOs), binds ZEB2 promoter, destabilize ZEB2 mRNA. As a result, it inhibit ZEB2‐induced EMT[57] (Figure 3).
General Comment: Page 1, Lines 36-38. PTMs are usually enzymatic because, for instance, co-factors binding are also considered PTMs. The authors should accommodate this information when explaining PTMs.
Reply: Thank you for your comment. We added some sentence covering your comments.
Red characters were used to highlight improved parts of the revised manuscript.
In addition, it is found through structural studies and biochemical studies that cofactors present through covalent bonds at the active sites of enzymes also undergo the PTM process [18-20].
General Comment: Page 3 lines 123-125. Several proteins are listed but not explained.
Reply: Thank you for your comment. We have revised the text according to your comments. Red characters were used to highlight improved parts of the revised manuscript.
The human SNAI1 is located on chromosome 20q13.13 and encodes the 264 amino acid Snail protein. It is a member of the Snail superfamily, acts as transcriptional regulators of EMT[78]. The human SNAI2 is located on chromosome 8q11.21 and encodes the 268 amino acid Slug protein. Slug binds the nuclear receptor corepressor (NCoR) and C-terminal binding protein 1 (CtBP1) to stabilize Slug and inhibit the expression of E-cadherin[79]. The human Twist1 genes is located on chromosome 7p21.2 and encodes the 202 amino acid Twist1 protein. the Twist1 plays a critical role in the progression of cancer by modulating EMT[80,81].
General Comment: Fig. 1: Very small with low resolution (compare to fig. 2). Top: the meaning of the legends are not explained. It is probably my copy but colors do not match.
Reply: Thank you for your comment. We have revised the text and Figure according to your comments. Red characters were used to highlight improved parts of the revised manuscript.
Figure 1. Overviews of ZEB1 and ZEB2 PTMs. It is characterized by the presence of two zinc finger clusters, one at each end (NZF and CZF) and located homeodomain (HD). Other domains are P300-P/CAF interaction domain (CBD) and the Smad binding domain (SBD) and the CtBP interaction domain (CID). ZF, zinc finger; NLS, nuclear localization signal. PTM site. Black, Phosphorylation (ZEB1, ZEB2); Sky blue, Ubiquitination (ZEB1, ZEB2); Red, SUMOylation (ZEB1, ZEB2); Green, Di-methylation (ZEB1, ZEB2); Orange, Acetylation (ZEB2).
General Comment: Page 5 lines 178-181 are in red.
Reply: Thank you for your comment. We changed characters color to black
General Comment: Wold it be possible to add a section with a good discussion about medical treatments, in tests or already being applied, that target PTMs in ZEBs?
Reply: Thank you for your comment. We have revised the text according to your comments. Red characters were used to highlight improved parts of the revised manuscript.
- Targeting ZEB modification for cancer therapy and therapeutic resistance.
ZEB is highly expressed in several cancers. Inhibiting the biological function of ZEB is a new approach in cancer therapy. The expression, stability, and localization of ZEB could interfere with different interactions in cancer patients. Phosphorylation of retinoblastoma (Rb) enhances the progression and metastasis of several cancers. The mechanism of RB in cancer is associated with the regulation of cyclins and cyclin dependent kinase. Rb dephosphorylation regulates EMT by inhibition the ZEB1 transcriptional activity in breast cancer[124].
General Comment: Table 1. Whats is the meaning of “Regulation factor”? It may be better explained in the text.
Reply: Thank you for your comment. We changed ‘Regulation factor’ to Kinase/Enzyme
General Comment: A ZEB1/2 amino acid alignment would be a nice addition to aid the reader follow the sites of PTMs.
Reply: Thank you for your comment. We have revised Figure 1 according to your comments.
Thank you so much for sharing your precious time with us. Our manuscript was significantly improved based on the constructive comments.

Reviewer 2 Report
The review by Park et al. on "Post-translational modification of ZEB family members in cancer progression" discussed the PTMs on ZEB proteins and how these PTMs regulates ZEB function during tumorigenesis. The review is well described; however, I have one minor suggestion.
1. Did the author found any paper that discussed the function of ZEB PTMs in cancer therapy resistance. If so, please add few lines on the role of ZEB PTMs in cancer therapy resistance.
Author Response
[To reviewer 2]
Dear Professor,
We are grateful for your consideration of our manuscript entitled “Post-translational modification of ZEB family members in cancer progression” (manuscript ID: ijms-2031734) by Park et al. and appreciate your helpful comments.
Replies to the comments are as follows:
General Comment: 1. Did the author found any paper that discussed the function of ZEB PTMs in cancer therapy resistance. If so, please add few lines on the role of ZEB PTMs in cancer therapy resistance.
Reply: Thank you for your comment. We have revised the text according to your comments. Red characters were used to highlight improved parts of the revised manuscript.
There are ZEB PTMs in resistance to various anticancer therapies[125]. Expression level of ZEB1 by induction ATM activation occurs chemoresistance in human breast cancer[126]. ZEB1 regulates tumor radioresistance by deubiquitylation of USP7. This in turn upreg-ulates homologous recombination-dependent DNA damage repair (DDR) and tumor radioresistance[14]. ZEB1 also modulates the radiotherapy resistance by the inhibition miR-205[127].
Thank you so much for sharing your precious time with us. Our manuscript was significantly improved based on the constructive comments.

Round 2
Reviewer 1 Report
The authors did a good job answering the criticisms. I just recommend minor revisions on style and spell check.